



# Carbon monoxide (CO) cycling in the Fram Strait, Arctic Ocean

Hanna I. Campen[1], Damian L. Arévalo-Martínez[1,2,3], Hermann W. Bange[1,*]

[1]Marine Biogeochemistry, GEOMAR Helmholtz Centre for Ocean Research Kiel, Düsternbrooker Weg 20, 24105 Kiel, Germany
[2]Institute of Geosciences, Kiel University, Kiel, Germany
[3]now at Radboud University, Nijmegen, The Netherlands

*Correspondence to*: Hermann W. Bange (hbange@geomar.de)

**Abstract.** Carbon monoxide (CO) influences the radiative budget and oxidative capacity of the atmosphere over the Arctic Ocean, which is a source of atmospheric CO. Yet, oceanic CO cycling is understudied in this area, particularly in view of the ongoing rapid environmental changes. We present results from incubation experiments conducted in the Fram Strait in August/September 2019 under different environmental conditions: While lower pH did not affect CO production ($GP_{CO}$) or consumption ($k_{CO}$) rates, enhanced $GP_{CO}$ and $k_{CO}$ were positively correlated with coloured dissolved organic matter (CDOM) and dissolved nitrate concentrations, respectively, suggesting microbial CO uptake under oligotrophic conditions to be a driving factor for variability in CO surface concentrations. Both production and consumption of CO will likely increase in the future, but it is unknown which process will dominate. Our results will help to improve models predicting future CO concentrations and emissions and their effects on the radiative budget and the oxidative capacity of the Arctic atmosphere.

## 1 Introduction

Carbon monoxide (CO) is a short-lived atmospheric trace gas which plays an important role for the radiative budget and oxidative capacity of the Earth's atmosphere (Forster et al., 2021). Overall, the surface ocean is a minor source of atmospheric CO contributing about 0.4 to 0.8 % to the natural and anthropogenic sources of CO (Conte et al., 2019; Zheng et al., 2019). However, CO has a comparably short atmospheric lifetime of 1 - 3 months (Zheng et al., 2019) and thus its oceanic emissions can contribute significantly to the atmospheric CO budget in the atmospheric boundary layer of remote areas such as the Arctic Ocean where the influence of other CO sources is marginal (Blomquist et al., 2012; Kort et al., 2012). However, there are only a few studies on dissolved CO in the Arctic Ocean (see e.g. Tran et al., 2013; Xie et al., 2009; Xie and Gosselin, 2005). In general the variability of dissolved CO concentrations is higher in the Arctic Ocean as compared to other ocean basins (e.g. Tran et al., 2013; Stubbins et al., 2006a, Park and Rhee, 2019). Particularly high CO concentrations were measured within bottom sea ice colonized by algae (Song et al., 2011; Xie and Gosselin, 2005).

Oceanic CO is mainly produced photochemically via the reaction of UV-light with coloured dissolved organic matter (CDOM) (see e.g., Ossala et al., 2022; Powers and Miller, 2015; Stubbins et al., 2006b; Wilson et al., 1970) and particulate organic





matter (POM) (see e.g., Song and Xie, 2017; Xie and Zafiriou, 2009). There is also evidence for thermal (dark) CO production from dissolved organic matter (DOM) (Zhang et al., 2008) and for biological CO production by phytoplankton (Gros et al., 2009; McLeod et al., 2021). Tran et al. (2013) suggested that *Phaeocystis sp.*, dinoflagellates and, to a lesser extent, diatoms are the major biological CO producers in the Fram Strait. However, the CO production by algae lacks research on the

physiological mechanisms and their interdependencies with biogeochemical parameters (Campen et al., 2021). Besides the emissions to the atmosphere, microbial consumption is the major loss process of CO in the ocean (Greening and Grinter, 2022; Conrad et al., 1982; Xie et al., 2005).

Ongoing environmental changes in the Arctic Ocean such as the loss of sea ice, changing light penetration in the upper ocean,

ocean acidification and altered nutrient and organic material supply (e.g. Thackeray and Hall, 2019; Hopwood et al., 2018; Stedmon et al., 2011; Terhaar et al., 2020) might affect CO production and consumption pathways as well as its emissions to the atmosphere from this region (Campen et al., 2021). The distribution and magnitude of coastal nutrient fluxes is predicted to change (e.g. Hopwood et al., 2018) due to increasing freshwater inputs via ice melting, which could lead to increased stratification and, in turn, limiting nutrient availability in the surface layer (Lannuzel et al., 2020). However, between 2012

and 2018 chlorophyll *a* concentration in Arctic Ocean surface waters increased 16 times faster than before, suggesting an additional input of nutrients that could fuel an increase in primary production (Ardyna and Arrigo, 2020) which in turn might lead to an increase in precursors of CO such as CDOM. Furthermore, light availability and penetration at the ocean surface is projected to increase due to loss of ice and decreasing albedo (Pistone et al., 2014; Castellani et al., 2022), potentially enhancing CO production in open surface waters and under-ice water during the melting season. Due to the increase of atmospheric

carbon dioxide ($CO_2$), the pH in the surface ocean is decreasing (Canadell et al., 2021) and model projections suggest that pH in Arctic Ocean surface waters could significantly decrease by the end of this century (Terhaar et al., 2020). Decreasing pH (i.e. ocean acidification, OA) is likely to influence the CDOM pool which, in turn, would alter CO production processes (Doney et al., 2009; Hopkins et al., 2020). However, to our knowledge, no studies on the effect of OA on CO cycling in the ocean have been published (Hopkins et al., 2020). How these environmental changes will affect CO production and emissions from the

Arctic Ocean is unknown so far due to limited measurements and knowledge gaps with regards to its sources and sinks. To this end, the major objectives of our study were (i) to identify the main drivers of CO production and consumption in the Fram Fram Strait and (ii) to assess the effect of ocean acidification on CO cycling.

## 2 Materials and Methods

### 2.1 Study area

The study was conducted on board the RSS James Clark Ross during the JR18007 cruise to the Fram Strait from 4 August to 6 September 2019. The Fram Strait, located between the west coast of Svalbard and the east coast of Greenland, is characterized by the inflow of Atlantic water via the West Spitzbergen Current (WSC) in the east and Arctic water outflow via the East



Greenland Current (EGC) in the west (Rudels et al., 2015; Marinela et al., 2016). Four incubation experiments were conducted
at stations NT6A, Ice2, D7 and D5 (Fig. 1). The stations NT6A, Ice2 and D5 were located at the shelf break, whereas D7 was
located in the open ocean region of the Fram Strait. Moreover, Ice 2 and D5 were in proximity to the ice edge. The EGC
affected Ice2 as indicated by its lower salinities and colder water temperatures, whereas D5 and D7 were influenced by warmer
and more saline Atlantic waters of the WSC (see Table S1 in the Supplement).

**2.2 Experimental setup**

For the incubation experiments, seawater from 5 m water depth was drawn from Niskin bottles attached to a 12-bottle
CTD/rosette and subsequently incubated in experimental enclosures for up to 48h. In total, eighteen 3.5 L light-transmitting
incubation bottles (DURAN®, quartz glass, GL 45, DWK Life Sciences, Germany) were filled with seawater. Lids (GL 45)
had Teflon-coated septa to easily press out the bulk water and close the bottles gas tightly. Teflon was chosen to minimize the
influence of plastic-derived CO in the experimental setup (Xie et al., 2002). Shading was minimized and natural light exposure
was maximized by placing the bottles upside down in the incubators, which were fixed on a mostly non-shaded area of the
ship's deck. To characterize the setting of the upper water a vertical profile down to 100 m was performed before the start of
the incubations. CO concentrations and ancillary measurements (see S2) from 5 m water depth served as sampling time 0 ($t_0$)
of the incubations.


Triplicate bubble-free seawater samples for the determination of dissolved CO were taken in 100 mL glass vials (both from
Niskin bottles and incubation bottles) by using a Tygon® tubing to avoid contamination by silicone rubber (Xie et al., 2002).
The vials were immediately sealed and stored between 0 and 6 °C in the dark to suppress further CO photoproduction. CDOM
was sampled in brown 500 mL glass vials with a screwed cap. Inorganic nitrate samples were drawn into 10 mL polyethylene
tubes, which were pre-rinsed three times with sample water and stored at -80° C until analysis at the Chemical Oceanography
Department of GEOMAR. CDOM samples were stored in the dark and below 5 °C until filtration (for method details see S2).
The pH in each experiment was manipulated to represent three different atmospheric $CO_2$ mole fractions: 405.43 ± 0.05
(Dlugokencky and Thoning, 2021), 670 and 936 ppm $CO_2$ for the treatments named ambient, pH1 and pH2, respectively. To
this end, the pH in pH1 and pH2, were adjusted by -0.14 and -0.3, respectively, to approximate the IPCC's representative
concentration pathway (RCP) 4.5 (moderate change) and RCP 8.5 (extreme change) relative to the ambient carbonate
chemistry of the seawater at the time of the sampling. To manipulate the carbonate system, $NaHCO_3$ and HCl were added
(Riebesell et al., 2011) and immediately checked for the resulting total alkalinity (TA) and dissolved inorganic (DIC)
concentrations. Values of $p$$CO_2$ and $pH_T$ (total scale) were calculated with the software CO2sys (Lewis and Wallace, 1998).
Immediately after pH manipulation, bottles were gas tightly closed and incubated.




Light incubators had transparent Plexiglas® sidewalls (GS 2458 UV transmitting) and no lid, so that the full natural sunlight spectrum could penetrate the enclosed incubation bottles from the sides and above (self-manufactured according to experimental needs, Fig. S3.1 in the supplements). While these incubators were placed on deck to allow natural sunlight penetration, black and covered water chambers served as dark incubators to exclude any light. All incubators were continuously

flushed with ambient seawater to keep bottles at ambient temperature. Light and temperature were monitored continuously in each incubator (HOBO pendant® temperature/light, onset, USA). Oxygen saturation (in %) was monitored to make sure that the incubations did not become anoxic (O2xyDot®, OxySense, USA). CO concentrations were determined at the beginning of the incubation ($t_0$), after 12 h ($t_{12}$), 24 h ($t_{24}$) and 48 h ($t_{48}$) of incubation (Fig. S3.1).

**2.3 CO measurements**

Dissolved CO concentrations were determined by the headspace method as described by Xie et al. (2002). We established a headspace by injecting 15 mL of CO-free synthetic air (purified via MicroTorr series, 906 media, SAES group, USA). The samples were then equilibrated for eight minutes (Law et al., 2002; Xiaolan et al., 2010). A 5 mL subsample from the equilibrated headspace was injected with a gastight syringe into the sample loop of a CO analyser (ta3000 AMETEK, USA).

Every sixth sample injection was followed by the injection of a standard gas mixture with 113.9 ppb CO in synthetic air (DEUSTE Gas Solutions, Germany) which was calibrated against a certified standard gas (250.5 ppb CO, calibrated against the NOAA 2004 scale at the Max Plank Institute for Biogeochemistry, Jena, Germany). This value was chosen as it lies in the expected range of the CO mole fraction equilibrated with open ocean waters.

Measured CO mole fractions from the headspace were corrected for the drift of the detector with the standard gas measurements and corrected for water vapour (Wiesenburg and Guinasso, 1979). The final dissolved CO concentrations were calculated based on Stubbins et al. (2006) with the solubility coefficients from Wiesenburg and Guinasso (1979). For each of the CO concentration triplicates we calculated the arithmetic mean and estimated the standard error according to David (1951). The overall mean error for the measurements of dissolved CO was ± 0.025 nmol L$^{-1}$ (± 17.4 %).


**2.4 CO consumption and production rates**

Net CO consumption ($NC_{CO}$) and net production rates ($NP_{CO}$) were calculated as the slope of the linear regression line for CO concentration [CO] loss and increase over the duration of the experiment (48 hours) and per pH treatment:

$$NC_{CO} = -[CO] * t^{-1} \qquad (1)$$

$$NP_{CO} = [CO] * t^{-1} \qquad (2)$$

Gross production rates of CO ($GP_{CO}$) were calculated as the sum of $NP_{CO}$ and the absolute value of $NC_{CO}$ in order to demask the effect of microbial CO consumption in the light experiments:

$$GP_{CO} = NP_{CO} + NC_{CO} \qquad (3)$$





To increase data points when possible, single CO gross production rates (singleGP$_{CO}$) were calculated between two sampling
times (0 – 12 h, 0 – 24 h, 0 – 48 h) for each treatment and for each experiment, respectively. Since consumption rates followed
a first order loss for all experiments (Fig. 2 and Fig. S3.2), the consumption rate constant ($k_{CO}$) for each experiment was
determined as the slope of the respective linear regression.

## 3 Results and Discussion

### 3.1 CO concentration development during dark and light incubations
The low initial CO concentrations (Table 1) are in line with the observation that CO in surface waters can show a pronounced
seasonal variability in Arctic waters. For example, Xie et al. (2009) reported considerably lower CO concentrations for
September/October 2003 (0.17 – 1.34 nmol L$^{-1}$) than for June 2004 (0.98 – 13 nmol L$^{-1}$) in the Amundsen Gulf (Beaufort Sea).
Tran et al. (2013) reported a mean CO concentration of 6.5 +/- 3.2 nmol L$^{-1}$ in Polar waters of the Fram Strait in July 2010.
And only recently Gros et al. (2022) reported mean CO concentrations in the range from 1.5 +/- 1.7 nmol L$^{-1}$ (in surface waters
at sea-ice stations) to 5.9 +/- 2.9 nmol L$^{-1}$ (in Polar waters) from the Fram Strait in May/June 2015.

CO concentrations decreased with time in all dark incubations, with the exception of pH2 at NT6A (Fig. 3 and Fig. S3.2).
While the general decrease in CO was most likely driven by microbial consumption which is the major known CO consumption
process in Arctic waters (e.g. Xie et al., 2005; Xie et al., 2009), elevated CO concentrations at NT6A (pH2) could hint towards
ongoing thermal CO production (Zhang et al., 2008). All light treatments showed a diurnal pattern of light intensity, though
light was never completely absent because the incubations were performed in the Arctic summer. CO concentrations in the
light incubations showed no uniform trend with time. Only during the incubations NT6A and D5 a significant increase of CO
concentrations over 48 h was observed. However, this net production which includes microbial CO consumption. Since there
was no obvious relationship between the timing of the sampling, CO concentrations and preceeding light intensities (Fig. 3),
this indicates that photochemical CO production did not exceed CO consumption. We therefore suggest that if there was
photochemical CO production, it was directly consumed by bacteria. Alternatively, biological CO production by phytoplankton
(Gros et al., 2009; Tran et al., 2013) or bacterioplankton and/or thermal production might have been dominant at NT6A and
D5 (Zhang et al., 2008).


The $k_{CO}$ computed from our experiments (Table 1) are comparable to previously published findings from Arctic waters: Xie et
al. (2005) reported first order consumption rates constants $k_{CO}$ of -0.040 ± -0.012 hr$^{-1}$ and -0.020 ± -0.0060 hr$^{-1}$ in the coastal
and offshore Beaufort Sea, respectively. (Please note that $k_{CO}$ are given as positive values in Xie et al. (2005)).

In general, a lower pH did not affect the CO concentrations neither in the dark incubations nor in the light incubations, since
the CO concentrations in the pH manipulated treatments did not differ significantly from the ambient treatments (as indicated



by the error bars in Fig. 3). Accordingly, pH affected neither $k_{CO}$ nor $GP_{CO}$ significantly during our incubations (see also Fig. S3.3).

**3.2 Effect of environmental variability on CO consumption and production**

We observed contrasting hydrographic settings at the stations selected or the incubation experiments. While Ice2 was located close to the ice edge and had a low water temperature and low salinity at $t_0$, D7 was located in the open Fram Strait with a higher water temperature and salinity at $t_0$ (Fig. 4). Therefore, Ice2 was most probably affected by freshwater input from ice melting and polar waters carried by the EGC (Fig. 4 and Table 1). D5 had a lower salinity compared to D7 and was also (at

least partly) affected by freshwater from ice melting. NT6A had a low salinity which was comparable to Ice2 but the water temperature at $t_0$ was much higher compared to Ice2. Moreover, station NT6A had a steep halocline in about 10 m, whereas Ice2 was well mixed in the upper layer (Fig. S3.4). Therefore, NT6A also being the southernmost station during our study had an apparently different hydrographic setting in comparison to the other three stations. When considering all stations except for NT6A, GPco showed a statistically significant correlation ($R^2 = 0.58$, $p < 0.05$) with increasing density. This suggests that

surface waters in the Fram Strait with a higher fraction of freshwater (i.e. lower density), due to e.g. fresh meltwater or polar inflow in the west Fram Strait, potentially lead to higher CO production rates. There was no significant relationship for $k_{CO}$ with density, which indicates that besides meltwater and polar waters additional factors must have influenced CO consumption in the area at the time of sampling.

Given that CDOM is the major driver for CO photoproduction in the ocean (see Introduction), a good correlation between both was an underlying assumption during our experiments. We observed that CDOM absorption for all treatments significantly correlated with singleGP$_{CO}$ ($R^2 = 0.45$, $p < 0.05$, Fig. 4; data from NT6A excluded). Moreover, CDOM absorption at $t_0$ was significantly correlated with $k_{CO}$ ($R^2 = 0.57$, $p < 0.05$). Given that photochemical production from CDOM is a CO source, this is most likely an indirect correlation: High CDOM absorption induces photochemical CO production which, in turn, results in

higher CO consumption (i.e. a lower $k_{CO}$) because $k_{CO}$ depends on the initial CO concentration.

Neither $GP_{CO}$ nor $k_{CO}$ was significantly correlated with Chl $a$ concentrations during our experiments (Fig. 4). This is in contrast to Xie at al. (2005) who reported a negative correlation between Chl $a$ and $k_{CO}$ (please note again that Xie et al. (2005) reported $k_{CO}$ as positive values). This suggests that Chl $a$ / $k_{CO}$ relationships seems to be variable within the Arctic realm, possibly as a

result of the complex interplay between different water masses (Cherkasheva et al., 2014; Rudels et al., 2015). Nitrate ($NO_3^-$) concentrations at $t_0$ and $GP_{CO}$ were negatively correlated (albeit statistically not significant at the 95% significance level and after excluding NT6A data; Fig. 4), while $k_{CO}$ was positively correlated with $NO_3^-$ concentrations at $t_0$ ($R^2 = 0.78$, $p<0.05$, Fig. 4). The combination of relatively higher Chl $a$ concentrations at $t_0$ with lower $NO_3^-$ concentrations at Ice2 and D7 with respect to NTA6 and D5, could explain the higher CO consumption rates at the two stations: On the one hand, CO is known to act as



competitive inhibitor for ammonium monooxygenase (*amoA*; the enzyme responsible for ammonium oxidation during nitrification; Zhang et al., 2020), resulting in cell uptake of CO under nutrient-deprived conditions (see Vanzella et al., 1989) as those found at the time of sampling. On the other hand, field and laboratory studies (Moran and Miller, 2007 and references therein; Cordero et al. (2019) have shown the ability of bacterioplankton (e.g. the *Roseobacter* clade) to oxidize CO during heterotrophic growth (i.e. using it as a supplementary energy source rather than a fixed carbon source for building biomass),

in particular under oligotrophic conditions. The fact that we still measured oxidation rates in waters with very low CO concentrations might indicate that the dominant community is rather heterotrophic, which in turn could help explaining the poor correlation with Chl *a*. This finding is important for modelling studies constraining marine CO sources and sinks in the framework of future scenarios where warming-derived stratification reduces $NO_3^-$ supply to the surface ocean. Under such 'starvation' conditions, inorganic compounds such as CO could help sustaining small planktonic communities.

Recent results show that $NO_3^-$ can enhance the photoproduction of carbonyl sulphide (OCS) (Li et al., 2022). OCS and CO photoproduction have a common intermediate in their photoproduction pathways, but photoproduction of OCS and CO in natural waters is anticorrelated (Pos et al., 1998). This might explain the trend of decreasing CO photoproduction ($GP_{co}$) with increasing $NO_3^-$ concentrations (see above).

## 4 Conclusions

In order to decipher the cycling of CO in the surface waters of the Fram Strait, we measured CO production and consumption rates in various incubation experiments at four sites in the Fram Strait in summer 2019. Our results show that lower pH (representing future scenarios of ocean acidification) did not affect CO gross production ($GP_{CO}$) and consumption ($k_{CO}$) rates. We observed a tight coupling of CO production and consumption. Hence, the produced CO is not necessarily emitted to the

atmosphere as the dissolved CO seems to be rapidly consumed before its atmospheric release. We therefore infer that CO consumption mainly drives dissolved CO concentrations and hence could act as a 'filter' for the subsequent atmospheric CO emissions from the Fram Strait. High rates of both CO production and CO consumption were favoured by a combination of high CDOM and low $NO_3^-$ concentrations. This suggests a photochemical production of CO from CDOM which, in turn, is consumed rapidly by microbes preferably under oligotrophic conditions such as those found at the time of sampling. In the

Arctic Ocean/Fram Strait, such conditions can be found, at least transiently, both at ice edges as well as in the open ocean where a supply of nutrients via melting and/or mixing is followed by stratification (Cherkasheva et al., 2014). We identified both CDOM and $NO_3^-$ as key drivers of CO cycling. This has the implication that predicted changes in terrestrial-derived and marine CDOM (e.g. Lannuzel et al., 2020), as well as dissolved $NO_3^-$ inputs (Tuerena et al., 2022) could affect future CO production and consumption in the region. Both trends might lead to higher CO gross production as well as higher CO

consumption. It is yet uncertain whether both terms will balance each other out (as observed in our study) or whether one process will become dominant. The question if and under which conditions CO consumption rates would stagnate should be addressed in future research, since in that situation CO would actually be emitted. Performing further multifactorial





experiments including i.e. UV light intensity and bacterial community data could help to elucidate the explanatory power of the different environmental factors on both CO production and consumption. This would facilitate a better incorporation of both terms into biogeochemical models, and would improve both CO emission estimates for the Arctic realm, and the assessment of how atmospheric CO emissions will affect the radiative budget and oxidative capacity of the Arctic atmosphere.

**Author contributions**

The study was conceived by HIC and HWB. HIC conducted the field work during the JR18007 cruise, and carried out the data analysis with HWB. DLAM supported the data analysis and interpretation as well as the preparation of the plots. HIC wrote the manuscript with contributions from HWB and DLAM.

**Competing interests**

The authors declare that they have no conflict of interest.

**Acknowledgements**

We thank the captain and crew of RSS James Clark Ross as well as the chief scientist David Pond for their support of our work at sea. We are grateful for the support of Tina Fiedler, Mehmet Can Köse, Zara Botterell, Patrick Downes, Oban Jones and Stephanie Sargeant during the cruise JR18007 and we thank Josephine Kretschmer, Nirma Kundu, Pratirupa Bardhan and Riel Carlo Ingeniero for their help with sample processing at GEOMAR. Moreover, we thank Yuri Artioli, Birthe Zäncker, Dennis Booge and Jonathan Wiskandt for helpful discussions on the data and programming support. This work contains data supplied by UK Research and Innovation - Natural Environment Research Council (UKRI-NERC) and is a contribution to the PETRA project (FKZ 03F0808A, NE/R012830/1) which was jointly funded by the German Ministry of Education and Research (BMBF) and UKRI-NERC as part of the Changing Arctic Ocean programme, see www.changing-arctic-ocean.ac.uk.

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



**Figures**

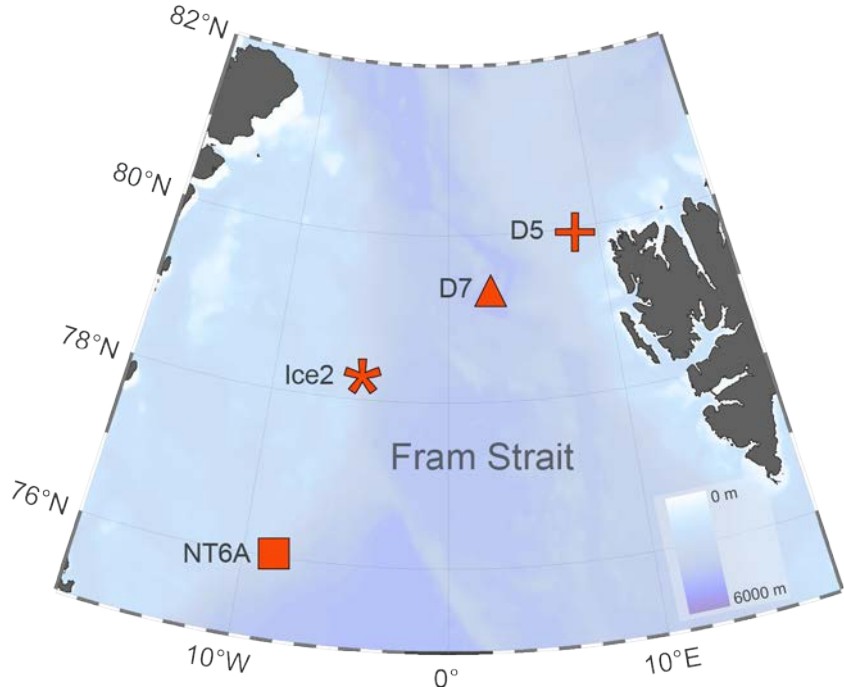


**Figure 1: Map showing the locations where incubation experiments were performed (stations NT6A, Ice2, D7 and D5).**

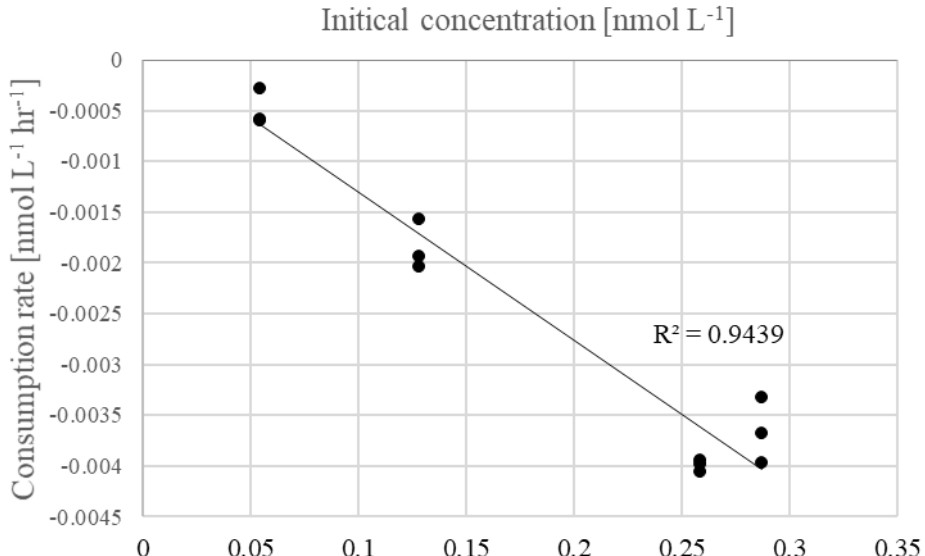

**Fig. 2 Initial CO concentrations plotted against overall consumption rates per experiment. All consumption rates depend on the**
**initial CO concentration (i.e. first order loss; $R^2 = 0.94$ with $p < 0.05$; see also Fig. S3.2).**

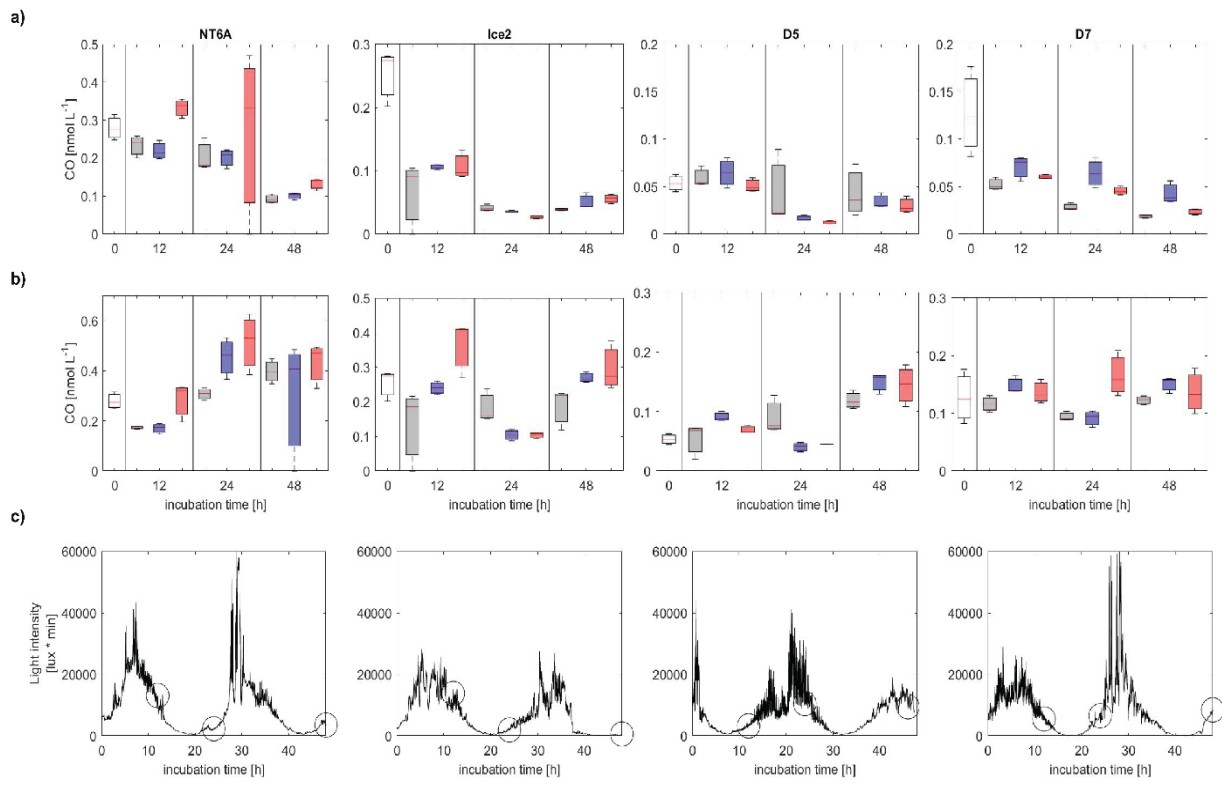

**Fig. 3 Development of CO concentration (nmol L⁻¹) over 48 hours of incubations a) in the dark and b) in natural sunlight. c) shows the respective light intensities in the light treatments at each station (light intensities in the dark treatment were zero). Circles indicate the timing of sampling events in dark and light treatments. white = initial concentration, grey = ambient, blue = pH1, red = pH2. The station names are indicated on the top. Please note that the scales of the y axes are varying between stations according to their CO maximum concentrations.**


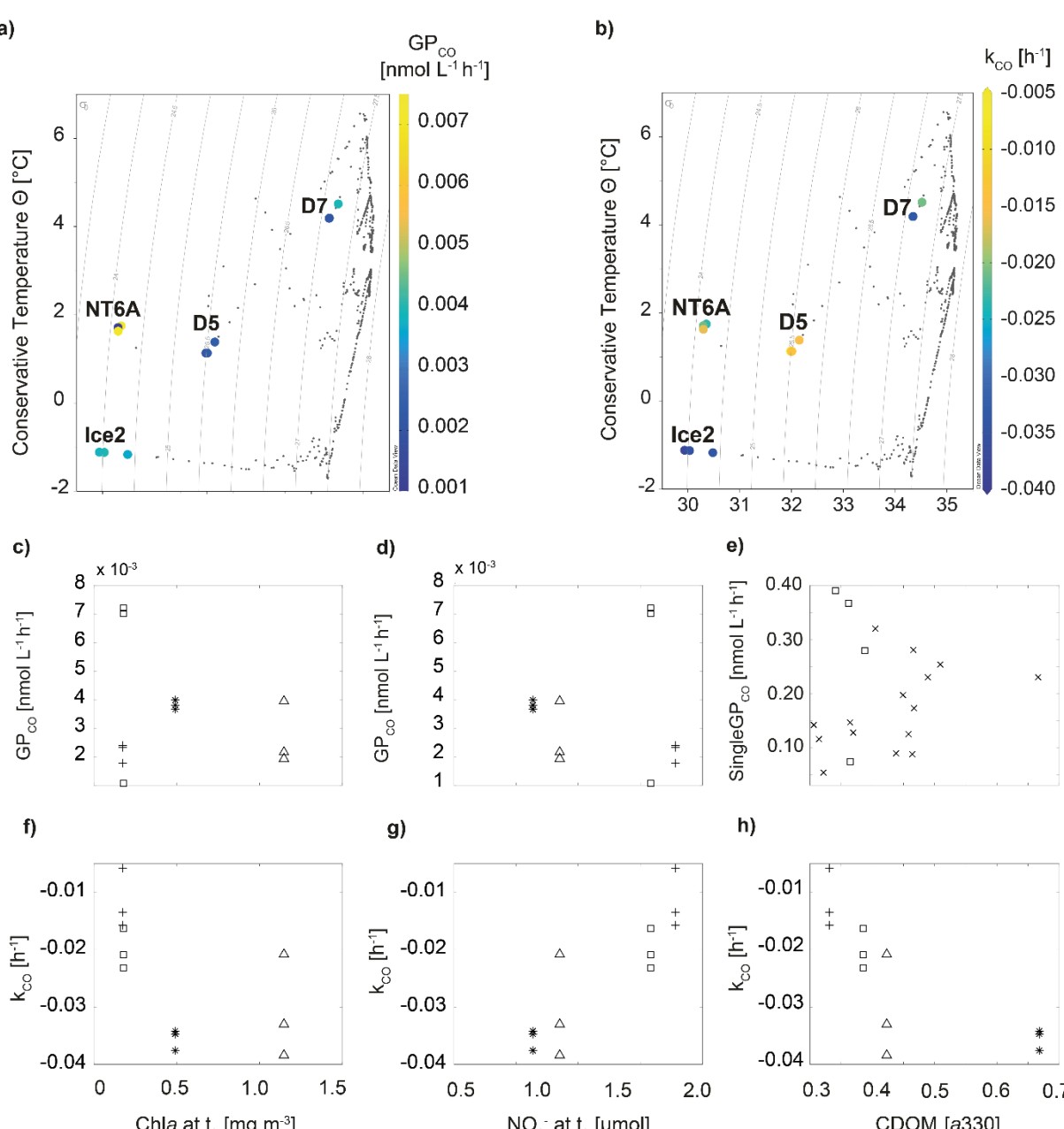

**Fig. 4 Relationship between GP$_{CO}$, $k_{CO}$ and selected environmental variables during the study. a) temperature/salinity plot including GP$_{CO}$, b) temperature/salinity plot including $k_{CO}$, c-d) GP$_{CO}$ vs. Chl$a$ and NO$_3^-$ at t$_0$, d) singleGP$_{CO}$ vs. CDOM absorption (330 nm) at each sampling time, f-h) $k_{CO}$ vs. Chl$a$, NO$_3^-$ and CDOM absorption (330 nm) at t$_0$. □ = NT6A, * = Ice2, + = D5, △ = D7, x = CDOM values at single sampling times of all stations excl. NT6A. In a) and b) isolines represent density.**





**Tables**

**Table 1: Initial CO concentrations and CO consumption rate constants ($k_{CO}$) of the four incubation experiments conducted at different pH levels. Data are given as mean ± estimate of standard deviation (for the initial CO concentrations) and as the slope of the linear regression ± error of the slope (for $k_{co}$).**


| Station | Initial CO concentration [nmol L$^{-1}$] | $k_{CO}$, amb [hr$^{-1}$] | $k_{CO}$, pH1 [hr$^{-1}$] | $k_{CO}$, pH2 [hr$^{-1}$] |
|---------|------------------------------------------|---------------------------|---------------------------|---------------------------|
| NT6A | 0.28 ± 0.035 | -0.023 ± 0.004 | -0.021 ± 0.003 | -0.016 ± 0.012 |
| Ice2 | 0.25 ± 0.041 | -0.038 ± 0.015 | -0.035 ± 0.018 | -0.034 ± 0.023 |
| D5 | 0.05 ± 0.009 | -0.006 ± 0.003 | -0.014 ± 0.019 | -0.016 ± 0.021 |
| D7 | 0.13 ± 0.049 | -0.038 ± 0.0095 | -0.021 ± 0.005 | -0.033 ± 0.005 |