# Peer review of "Carbon monoxide (CO) cycling in the Fram Strait, Arctic Ocean"

_Biogeosciences, 2022_

## Author Response (AR1)

Dear Tina,

thanks a lot for your comments. You wrote 'Please provide a short response to the editor (to me) about how you implemented the answer regarding the detection limit and the blanks in the revision.'

Here are our replies:

1) Detection limit: The detection limit was < 0.01 nmol L-1. Please note that the detection limit is so low that both CO concentrations and rate measurements were not affected.
   - We added (see lines 123-125): 'The lower detection limit of the CO analyser is 10 ppb CO in air which translates to a detection limit of about 0.01 nmol $L^{-1}$ for dissolved CO concentrations at equilibrium at water temperatures of -1 to 4 °C and salinities of 30 to 35.'
2) Blanks: We closely followed the recommendations for minimizing and avoiding CO contaminations given in Xie et al. (Mar Chem, 2002).
   - We added (see line 83): 'The vials were immediately sealed with Teflon-coated stoppers to minimize CO contamination (Xie et al., 2002).'
   - Blank measurements have been performed in the laboratory and during the campaign. We added (see lines 115-117): 'Blank measurements were performed before sample measurements by injecting CO-free synthetic air. No contamination by CO was detectable and, therefore, no blank correction was applied.'

We are looking forward to your reply and hope that the manuscript is now acceptable for publication.

With best regards

Hermann